# The Stability of Hop (*Humulus lupulus* L.) Resins during Long-Period Storage

**DOI:** 10.3390/plants12040936

**Published:** 2023-02-18

**Authors:** Ksenija Rutnik, Miha Ocvirk, Iztok Jože Košir

**Affiliations:** 1Department for Agrochemistry and Brewing, Slovenian Institute of Hop Research and Brewing, 3310 Žalec, Slovenia; 2Biotechnical Faculty, University of Ljubljana, 1000 Ljubljana, Slovenia

**Keywords:** alpha-acids, beta-acids, hop storage index, stability, hop storage

## Abstract

The stability of alpha-acids, beta-acids and hop storage index (HSI) values under different conditions (aerobic/anaerobic, 4 °C/room temperature) was studied in a two-year trial. Six different varieties (Celeia, Aurora, Bobek, Styrian Gold, Savinjski Golding and Styrian Wolf) were used in the form of cones and pellets. Alpha- and beta-acids were determined by HPLC and HSI by spectrophotometry. Anaerobic conditions at 4 °C were best for alpha-acids, beta-acids and HSI values; however, 10–35% of the alpha-acids were still lost after two years. The decline was greater (63–99%) under aerobic conditions and at room temperature. Alpha-/beta-acid ratios increased in hop cones and decreased in hop pellets, whereas HSI values increased in all storage conditions. Overall, the performance was better for pellets than for hop cones. Storage conditions, storage form and hop variety had significant effects on the stability of hop resins.

## 1. Introduction

The female inflorescences of the hop plant (*Humulus lupulus* L.), a perennial climbing plant, play an indispensable role in the brewing industry. These inflorescences contain lupulin, a complex of resins and essential oil [1]. The resins are further divided into soft and hard resins according to their solubility in hexane. Soft resins contain alpha- and beta-acids, the most valuable brewing compounds, as well as essential oils [2,3]. Alpha-acids, also referred to as humulones, are mixtures of five analogues: co-, n-, ad-, pre- and post-humulone, with last two being less common. They are insoluble in their native form, but they undergo isomerisation during the process of wort boiling, resulting in six soluble isomers (cis- and trans- forms of co-, n- and ad- analogue) that are responsible for 90–95% of beer bitterness [4]. In addition to the alpha-acid distribution, the ratio between co-humulone and (n+ad)-humulone is also an important contributor to beer bitterness. In 1972, Rigsby revealed that hops with high co-humulone content imparted stronger, sharper and less pleasant bitterness compared to low co-humulone hops [5]. His findings attracted substantial attention and had significant consequences for the hop and brewing industries. Brewers demanded more low co-humulone hops, which then led to breeding the low co-humulone varieties. This prompted other researchers, such as Wackerbauer and Balzer [6], Hughes [7], Shellhammer [8], and Schönberger [9], to investigate the influence of co-humulone levels on bitterness quality. However, they found no evidence for co-humulone causing a harsher or more unpleasant bitterness than observed with the other homologues.

Despite these findings, a belief remains among some brewers that high levels of co-humulone result in low-quality, bitter beers. In fact, hops with high co-humulone levels could contribute more bitterness, since the isomerisation yield is higher for co-humulone than for other homologues [9,10]. Besides undergoing isomerisation, alpha-acids also oxidise into humulinones, which are more polar than iso-alpha-acids due to the additional -OH group. Similarly, beta-acids are also the sum of their analogues, namely co-, n-, ad-, pre- and post-lupulone, with the last two being less common. The beta-acids cannot undergo isomerisation, as they lack a tertiary -OH group in the aromatic ring. On their own, their contribution to beer bitterness is nine times less than that of alpha-acids [11]. 

Losses of hop acids and essential oil begin as soon as the hops are harvested, with the rate of loss depending on various factors, such as hop variety, hop form, storage conditions, and time of storage. Knowledge of the extent of these losses is especially important when a hop surplus occurs on the market and afterwards when hops are scarce and breweries buy and use hops that can have been stored for as long as two years. To avoid using old hops, hop buyers are increasingly purchasing hops with hop storage index (HSI) values well below the desired levels. The HSI value is the indicator of hop freshness and it represents the ratio between oxidised and non-oxidised hop acids. Based on the HSI value, hops can be sorted into different categories [12]. The first category is fresh, with HSI values up to 0.32. The slightly aged category contains hops with HSI values between 0.33 and 0.40. Hops with HSI values from 0.41 to 0.50 fall into the aged category. Strongly aged hops are those with HSI values between 0.51 and 0.60, and hops with HSI values over 0.61 are over-aged. 

One of the most thorough experiments on hop ageing was done in 2012 by Mikyška and Krofta [13], who conducted a one-year experiment with T90 pellets of four different varieties. They found that anaerobic conditions at 2 °C for one year caused insignificant changes in the resin content. By contrast, anaerobic conditions at 20 °C resulted in a 20–25% loss of alpha-acids, whereas the beta-acids were stable at those conditions. The greatest decline was observed under aerobic conditions at 20 °C, where 64–88% of the alpha-acids and 51–83% of the beta-acids were lost. Srečec et al. [14] investigated the storage stability of pellets of the Aurora hop variety and found that after one year of storage, the pellets lost only 2% of their alpha-acids. Based on their HSI value (HSI 0.33–0.40), the stored Aurora pellets fell into the category of slightly aged. The Aurora pellets also fell into the same category if stored for six months under aerobic conditions at low temperatures. By contrast, when stored in air at 21 °C, the Aurora pellets lost 54% of their alpha-aids and their HSI value increased to 0.59. Tembo [15] investigated the effect of temperature and package atmospheric conditions on alpha- and beta-acid degradation in Cascade and Chinook hops after six months of storage. The rate of alpha-acid degradation increased with increasing temperature, whereas the beta-acids were more stable and showed no statistically significant difference after six months, even when the storage temperatures were as high as 25 and 35 °C. Tembo also determined that, at high temperatures, a nitrogen atmosphere performed better than an air atmosphere. Canbaş et al. [16] investigated the effect of temperature and anaerobic conditions on the chemical composition of hop pellets after six months of storage. They found that the pellets lost from 6.9 to 19.4% of their alpha-acids (variety dependent) at 3 °C and 30.8–36.4% if stored at room temperature. Unlike the findings reported by Tembo, Canbaş et al. found that beta-acids were not particularly stable, as the losses ranged from 3.2 to 6.3% at 3 °C, and from 10.7 to 22.7% at room temperature. In contrast to these studies, chemometric studies of hop degradation under different storage conditions [17] revealed that the presence of oxygen did not play a significant role in alpha-acid deterioration at low temperatures (≤−10 °C), whereas storage at aerobic conditions and at 27 °C led to complete deterioration of co-, n-, ad-humulone after six months of storage. Táborský et al. [18] investigated pellets of eight different varieties after storage at 4 °C under anaerobic conditions and found once again a variety-dependent loss of alpha- and beta-acids and a decrease in the alpha/beta-acid ratio in all varieties by the end of a nine-month storage experiment. 

The aim of our research was to extend the current knowledge about the stability of hop acids by conducting an experiment with six different hop varieties in the form of cones and pellets and storing them under four different storage conditions for two years. To our knowledge, this is the first study to compare the stability of hop resins in cones and pellets stored for two years under aerobic and anaerobic conditions at two different storage temperatures at the same site.

## 2. Results and Discussion

### 2.1. Alpha-Acids

The content of alpha-acids in hops is the most important beer brewing parameter for two reasons. First, the price of the hops is based on the alpha-acid content and second, the doses of hop in beer recipes are based on the weight (mg) of alpha-acids per litre and not on the weight (mg) of hops added per litre. A huge amount of alpha-acids can be lost during storage under improper conditions. Figure 1a–j shows the changes in alpha-acid content for the different hop varieties and shapes (cones or pellets). The conditions are indicated by the following letters: A-anaerobic, cold room; B-anaerobic, room temperature; C-aerobic, cold room; and D-aerobic, room temperature. The differences between different storage conditions were evaluated by performing one-way ANOVA at the 0.05 significance level for each variety and each form of hops. The Tukey’s means comparison test was subsequently performed to evaluate which groups showed differences and at which time points.

For the Celeia cones (Figure 1a), condition D caused a significant reduction in the alpha-acid content compared to the other conditions. The difference between condition D and the other three storage conditions was not statistically significant for the first six months (F(2,18) = [0.25], *p* = 0.78), but after six months, the storage was better under condition A than under conditions B (*p* = 0.001) or C (*p* < 0.001). The difference in alpha-acid content in Celeia cones stored under conditions B and C was not statistically significant, even after one year. However, as shown in Figure 1a, storage under condition B resulted in a slightly higher alpha-acid content. Celeia pellets (Figure 1b) had the highest alpha-acid content after two years if stored under condition A. Statistically, condition A was better than conditions C (*p* = 0.05) or D (*p* < 0.001), and condition D was worse than conditions B (*p* = 0.002) and C (*p* = 0.01). No difference was observed between conditions A and B until eight months of storage. The decline was subsequently more rapid under condition B and was comparable to that observed under condition D. The greatest loss of alpha-acids in both cones and pellets occurred within the first six months. The only exception was Celeia pellets stored under condition D, where the greatest loss occurred during the second half of the first year. 

For the Aurora variety, storage under condition D resulted in the lowest alpha-acid content, as shown in Figure 1c. No differences were noted between conditions A and B for the first seven months. Storage under condition C was worse than under conditions A (*p* = 0.003) and B (*p* = 0.02) in the first seven months. Subsequently, storage was statistically better under condition A than under condition B (*p* < 0.001). No statistical difference was detected between conditions B and C after month seven. As with the Celeia pellets, the Aurora pellets also retained the highest amount of alpha-acid if stored under condition A. 

In the first six months, no statistically significant differences were detected between conditions A and B or between conditions B and C; however, as shown in Figure 1d, hops stored under condition A retained a higher alpha-acid content compared to other conditions. After six months, differences were evident between all storage conditions ((F(3.44) = [28.90], *p* < 0.001), except between conditions B and C. After one year, the differences began to favour condition B. The worst condition was condition D, which showed alpha-acid losses of approximately 65% after two years. Comparison of the cones and pellets indicated that the pellets performed better under condition D (more than a 10% difference) and condition B (more than 5% difference), whereas the difference between conditions A and C was less than 5%; however, the difference between cones and pellets was not statistically significant.

The alpha-acid content of Bobek variety is shown in Figure 1e. The differences between the storage conditions were very similar to those observed for the Aurora cones, and storage under condition D was worse than under all other conditions (*p* < 0.001). Storage was also better under condition A than under condition C (*p* = 0.02). Comparison of conditions A and B showed that A performed better (*p* < 0.001). Bobek cones retained almost the same amount of alpha-acid for the first year, and condition B samples subsequently performed slightly better than condition C cones (*p* = 0.05). Bobek pellets (Figure 1f) stored under condition D performed the worst among all varieties, as more than 98% of the alpha-acids were lost in two years. Condition A was better than conditions B and C (*p* < 0.001), and no statistical difference was detected between conditions B and C. Interestingly, after one year, a slightly higher content was retained if C conditions were applied.

Figure 1g,h show the changes in the Styrian Wolf cones (Figure 1i) and pellets (Figure 1j). As with all the other varieties, the cones and pellets of Styrian Wolf lost the greatest amount of alpha-acids when stored under condition D. Styrian Wolf cones and pellets were the only samples that showed significant differences between conditions B and C from the beginning of the experiment (*p* = 0.009 for cones and *p* = 0.02 for pellets). Comparison of conditions A and B conditions again confirmed that A was better (*p* = 0.01 for cones and *p* < 0.001 for pellets). Pellets performed better under conditions A and B and no difference was evident between cones and pellets under condition C. Under condition D, the cones of Styrian Wolf performed better than the pellets.

For the Styrian Gold samples (Figure 1i), the differences for conditions A, B and C were smaller than in the other varieties, but condition A was again better than conditions B and C and condition D was the worst (*p* < 0.001). No difference was noted between conditions B and C. The pellets of Savinjski Golding (Figure 1j) showed similar patterns for the best and worst storage conditions, and the trend seen in Bobek pellets, where condition B was worse than condition C after two years, was statistically significant in the Savinjski pellets (*p* = 0.02). Interestingly, this difference was only observed with the Bobek and Savinjski Golding pellets.

Table 1 shows a summary of the losses after two years. A two-way ANOVA performed to analyse the effect of hop variety and storage conditions on alpha-acid loss after two years showed that both storage condition and variety and interaction between them significantly affected the loss of alpha-acids (F(12,20) = [3,24], *p* = 0.01). Further research (Tukey’s mean comparison) revealed that after two years all storage conditions were statistically different, with *p* values below 0.001 and 0.01 for the B–C interaction. Comparison of the different varieties indicated that only Aurora lost a significantly lower amount of alpha-acids. However, the time at which the less ideal storage conditions had significant impacts on alpha-acid content did differ among the varieties, confirming that the variety does play an important role in alpha-acid loss.

The differences in alpha-acid losses between cones and pellets were pronounced (>10%) only in rare cases, such as Aurora under condition D, Bobek under condition D, and Styrian Wolf under conditions A and D. This suggests that access to oxygen and higher temperatures affect cones and pellets somewhat differently. Overall, the shape did not have a statistically significant effect on alpha-acid loss. The greatest amount of alpha-acid was retained under anaerobic conditions in the cold room, in accordance with previous studies [13,14,15,16,17,18,19]. Mikyška and Krofta [13] found no difference in alpha-acid content after one year under anaerobic conditions at 2 °C, which is not consistent with our results, where pellets lost between 8.5 and 22.2% (after one year). Interestingly, on average, our varieties lost slightly more alpha-acids under condition B and somewhat less under condition D, if compared with the Czech varieties.

Comparison of the results by Srečec et al. [14] for Aurora allows the conclusion that pellets stored under inert gas (N_2_) lose even less alpha-acid content than those stored under vacuum. Losses under other conditions after six months and one year were similar to those of the present study. Verissimo [17] showed that presence/absence of atmospheric air was not statistically important at very low temperatures (below −10 °C), whereas in our study, the presence of oxygen was an important factor at low temperatures (4 °C). This finding indicates that the presence of oxygen is no longer important only at temperatures below the freezing point.

Alpha-acid content is predominantly the sum of co-humulone, n-humulone and ad-humulone. In our HPLC analysis, the n- and ad- analogues are eluted together; consequently, the data for them were combined. Table 2 presents the contents of co-humulone and (n+ad)-humulone for Aurora and Styrian Wolf cones and pellets at 0, 6, 12, 18 and 24 months. The last column shows the co-humulone content (%) for the alpha-acids. The results for all varieties are shown in Appendix A for cones and in Appendix A for pellets. As expected, the levels of co- and (n+ad)-humulones declined during storage. In all varieties, except Bobek, the loss of alpha-acid analogues was higher in cones than in pellets; however, the difference was not statistically significant at the 0.05 level. Another important parameter in hops is the ratio of co-humulone in the total alpha-acids, and this feature is variety-dependent. During the ripening phase, the percentage of co-humulone in alpha-acids increases and stabilises before the hops are fully ripe. In the present study, the ratios of individual analogues of humulone did not change, even after storage. The only exception was the Bobek pellets in the last period of the experiment under condition D, which we believe reflects the small quantities of homologues in the sample. One-way ANOVA revealed no statistical difference in the percentage of co-humulone in the total alpha-acids during storage in any variety. These results align well with previous studies that have shown no significant changes in the ratio of co-humulone in alpha-acids [13,20]. The observation by Krofta et al. [13] of a slight decrease in the co-humulone content in alpha-acids in hops stored under aerobic conditions was not confirmed in our study. This ratio could therefore be a useful hop variety indicator, even when hops are stored for a long time.

### 2.2. Beta-Acids

As observed for the alpha-acids, the beta-acid content is also the sum of co- and (n+ad)-humulone. The results of beta-acids storage trials are shown in Figure 2a–j. Figure 2a shows differences for Celeia cones, Figure 2b for Celeia pellets, Figure 2c for Aurora cones, Figure 2d for Aurora pellets, Figure 2e for Bobek cones, Figure 2f for Bobek pellets, Figure 2g for Styrian Wolf cones, Figure 2h for Styrian Wolf pellets, Figure 2i for Styrian Gold cones and Figure 2j for Savinjski golding pellets. Unlike the case for the alpha-acids, the form of the hop was a significant factor in the stability of beta-acids. Taking into account all varieties, all conditions and both shapes, higher amounts of beta-acids were retained in hop pellets than in hop cones (*p* < 0.001). This is because pelletising reduces the surface area of the hop samples, so they oxidise more slowly than hop cones. The only exception was the Bobek variety (Figure 2e,f), where complete degradation of both alpha- and beta-acids occurred in hop pellets under condition D. Anaerobic conditions in a cold room were favourable for retention of the greatest amounts of beta-acids in all hop varieties and sample shapes. Under condition A, the Aurora variety showed the greatest stability among the hop cones (Figure 2d), and the Styrian Wolf variety showed the best stability among the pellets (Figure 2h). The beta-acid losses ranged from 25–35% for hop cones and 11–33% for pellets after two years of storage. Comparison of conditions B and C revealed that the losses after two years were 10–50% higher for hops stored under condition C. However, Styrian Gold cones, Bobek pellets and Savinjski Golding pellets did not show significant differences when stored in the B and C conditions. Nevertheless, the atmospheric conditions had a higher impact on the loss of beta-acids than the temperature did (*p* < 0.01), thereby confirming the findings of Krofta et al. [20]. Under condition D, all varieties stored as hop cones lost more than 88% of their beta-acids. In pellets, the loss was more variety-dependent, as losses of beta-acids were 45% for Aurora, 60% for Savinjski Golding, 74% for Celeia, 93% for Styrian Wolf and 98% for Bobek. Interestingly, the Styrian Wolf pellets were the most stable under condition A but almost the least stable under conditions C and D. This is one more indicator that storage conditions do play an important role in hop stability.

In 2018, Táborský [18] conducted a nine-month storage trial on eight different varieties under condition A. Five of the eight varieties lost less than 3% of their beta-acids, two lost between 7 and 9% and one lost 31.3%. In a comparable time, our pellets under comparable conditions lost between 5 and 20%, which falls within the same range as reported for the Sládek (Czech R.), Tradition (Germany) and Galaxy (Australia) varieties. Tembo [15] found that the beta-acids remained at the same level for six months under condition A, in contrast to our results. These deviations could reflect the use of different varieties. The losses of beta-acids after six months reported by Canbaş [16] are in agreement with our results. Mikyška and Krofta [13] observed a loss of only up to 2% under anaerobic conditions, regardless of the temperature. Our findings do not seem to confirm this high stability of beta-acids under anaerobic conditions. However, the results for samples stored under condition D are comparable to ours. Appendix A show the contents of co-lupulone, (n+ad)-lupulone and the percentage of co-lupulone in beta-acids for months 0, 6, 12, 18 and 24. As expected, the contents of individual analogues decreased, with the frequency of the loss dependent on the variety, condition and sample shape. The percentage of co-humulone in alpha-acids did not change during storage, but some increases were noted in the percentage of co-lupulone in beta-acids. Increases in the percentage of co-humulone in beta-acids mean that the amounts of (n+ad)-lupulone are decreasing faster than those of co-humulone. In the cones, the differences were not statistically significant (*p* > 0.05), except for Styrian Wolf, where storage conditions did have an impact on the percentage of co-lupulone in beta-acids (*p* = 0.03), whereas storage time had no significant effect (*p* = 0.18). The pellets showed a statistically significant difference in the percentage of co-lupulone in beta-acids under condition D for the Celeia, Bobek and Styrian Wolf varieties. The Aurora and Savinjski Golding varieties showed an increase in the ratio, but it was not statistically significant. Additional studies are needed to confirm a faster decrease in co-lupulone than in (n+ad)-humulone in the pellets.

Beta-acids are known to deliver a 9-fold lower bitterness aspect to beer compared to alpha-acids [21]. However, they are very prone to oxidation reactions if exposed to oxygen [22], and this leads to the formation of eight degradation products, with hulupones as the major product. Hulupones provide a short, lingering, fine bitterness, similar to that of iso-alpha-acids but with less intensity. Algazzali and Shellhammer [23] reported that hulupones were 84% as bitter as iso-alpha-acids, which was greater than the value previously posted by Krofta and Mikyška [24]. The other beta-acid oxidation products leave a more unpleasant bitterness [25]. Analogues of hulupones are then further degraded into non-bittering hulupinic acid [25,26]. Humulinones, alpha-acids oxidation products, are 34% less bitter than iso-alpha-acids, but their higher solubility makes them more easily transferred into beer [11]. Ultimately, the ratio between alpha- and beta-acids becomes another important parameter in the formation of beer bitterness by hops. Aromatic hops have ratios around 1, while bittering hops possess higher ratios [27,28].

Therefore, in hops with a low alpha/beta ratio, the contribution of beta-acids to bitterness is no longer negligible. If the levels of alpha- and beta-acids decrease at the same rate during the storage trial, the ratios would remain the same. In previous studies [13,14,15,16,24], beta-acids were determined to be more stable than alpha-acids. Our results do not appear to completely corroborate this observation. Table 3 shows the ratios between alpha- and beta-acids at the beginning and the end of our trial.

Under condition A, the ratios remained essentially the same, suggesting that the deterioration of alpha- and beta-acids occurs at the same rate. Exceptions are the high-alpha-acid hop cones, where the ratio increases, meaning that beta-acids are decreasing faster than alpha-acids. Under conditions B, C and D, the hop form appears to play considerable role, since the beta-acids tend to decrease more rapidly in the hop cones, whereas the alpha-acids tend to decrease faster in the hop pellets. This is probably a consequence of the chemical properties of alpha- and beta-acids, as beta-acids are more prone to oxidation and pelletising the hops slows down the oxidation process due to the smaller surface area of the hop pellets. Oxidation is a particular problem in high-alpha–acid hops with high alpha/beta ratios, where the loss of alpha-acids cannot be compensated by the bitterness of beta-acid oxidation products.

### 2.3. Hop Storage Index

The HSI serves as a quality indicator, since it reflects the ratio between the levels of the oxidation products of acids and the acids themselves. It cannot be used to determine the exact age of a hop sample, but it can serve as a quick determinator of the proper handling and storage of hops. Figure 3a–j shows the percentage increase in the HSI values for all six varieties. Figure 3a shows differences between various conditions for Celeia cones, Figure 3b for Celeia pellets, Figure 3c for Aurora cones, Figure 3d for Aurora pellets, Figure 3e for Bobek cones, Figure 3f for Bobek pellets, Figure 3g for Styrian Wolf cones, Figure 3h for Styrian Wolf pellets, Figure 3i for Styrian Gold cones and Figure 3j for Savinjski golding pellets. Three-way ANOVA of all results (i.e., the full two-year trial) was performed to detect any statistical differences between conditions, varieties and forms. The population means of the interaction of all three factors were significantly different (*p* < 0.02), indicating that all three factors had an impact on the HSI value. A closer look (Tukey test) into the conditions revealed no statistical differences between conditions B and C after two years when all varieties were considered. Overall, the storage performance was better for hop pellets than for hop cones, especially under condition D. Under condition A, the smallest increase (13%) in the HSI was noted for Bobek pellets, which is interesting since the content of alpha- and beta-acids in Bobek was not very stable. Conversely, the Bobek pellets had the worst HSI under condition D for both cones and pellets. Under condition D, the hop cone HSI value was 400% of the starting value or higher. The cones of Celeia showed the highest increase in HSI (53%) under condition A. Conditions B and C had almost the same impact on HSI (50–100% increase), except for Styrian Wolf, where condition B was better than condition C. The most noticeable changes between conditions A and B/C were observed after 4–8 months (depending on the variety). After one year of storage under condition A, the HSI of the pellets increased by 8–15%, in agreement with the findings of Mikyška and Krofta [13], who found an increase of 5–16% under the same conditions. A similar effect was noted for condition B, where the percentage loss fluctuated at around 35% after one year. However, the increase in HSI under condition D was comparable in the Czech varieties and in the Bobek and Styrian Wolf pellets, but higher than in the Celeia, Aurora and Savinjski Golding pellets. Our results are also comparable with those reported by Tembo [15], who showed losses of about 5–15% after half a year under condition A. Rutnik et al., who investigated the impact of hop freshness on dry-hopped beer, determined that Aurora and Celeia produce a lower quality beer if hops with HSI values lower than 0.4 are used and that this limit is 0.5 for Styrian Wolf hops. If starting with hops with HSI values of 0.3, which is considered the upper limit for fresh and well-handled hops, all five varieties of pellets would not exceed an HSI value of 0.4 in two years if stored under condition A. The cone value would be somewhere around 0.4–0.45. Under condition D, those values would be reached in 2–8 months, depending on the variety and form.

Since the HSI value depends on the loss of alpha- and beta-acids, we also checked the correlation coefficient between them (Table 4). Almost all the R^2^ values were higher than 0.9, which confirms the correlation between HSI and the losses of alpha- and beta-acids. A few R^2^ values were slightly lower, especially in cases in which HSI did not change for some time.

## 3. Materials and Methods

### 3.1. Materials 

Methanol (HPLC grade) was purchased from J. T. Baker (Deventer, The Netherlands); toluene, diethyl ether, phosphoric acid (85%) and hydrochloric acid (37%) were obtained from Honeywell (Charlotte, NC, USA); sodium hydroxide was purchased from Sigma Aldrich (St. Louis, MO, USA); international calibration extract 4 (ICE4) was obtained from Labor Veritas (Zürich, Switzerland). Hop material was supplied by Hmezad exim d.d., except for the Bobek cones, which were supplied by Farm Bosnar and the Styrian Wolf cones, which were supplied by Farm Bizjak, both located in the Savinja valley, Slovenia. Four different hop varieties (Celeia, Aurora, Bobek, and Styrian Wolf) were used in the experiment in the form of both cones and pellets, whereas Styrian Gold was used only in cone form and Savinjski Golding was used only as pellets. All used varieties were grown in Slovenia.

### 3.2. Storage and Sampling

Each variety and each form of hops was weighed in 140 g lots and packed into aluminium bags (PET/Al/LDPE, 12/8/10 μm, with the matte Al side outside). Four different types of packages were prepared for each variety and each form: two vacuum sealed and two non-vacuum sealed. One of each package type was stored in a cold room (4 °C), and the other two were left at ambient temperature in a warehouse (where unsold hops are stored throughout the year when the use of a cold room is not possible). Enough samples were prepared to allow monthly analysis for two years. In total, 384 samples were packed and stored under the desired conditions. In the first year of the experiment, all samples (40 samples; each variety, each form, each storage temperature, and each packaging condition) were analysed monthly; in the second year, samples were analysed every two months. All samples were analysed in duplicate.

### 3.3. Alpha- and Beta-Acids Determination by HPLC

The contents of alpha- and beta-acids in the hop samples were determined following the procedure described in the official Analytica-EBC method 7.7 [29]. Briefly, 5 g of the hop sample was weighed into a flask. Ten millilitres of methanol, 20 mL of 0.1 N hydrochloric acid and 50 mL of diethyl ether were added, and the flask was attached to a shaker and shaken for 45 min. Five millilitres of upper phase was removed and dissolved in methanol. The samples were filtered into vials through 0.45 μm PTFE filters. The extracted compounds were separated on a Nucleodur ^®^ 5–100 C18 HPLC column (125 × 4 mm; Macherey-Nagel, Düren, Germany) using a mobile phase of methanol, distilled water and phosphoric acid (775:210:9; *v*/*v*/*v*). The flow rate was 1 mL/min, the thermostat was set at 40 °C and 2 µL of sample was injected. The compounds were detected using a diode array detector at 314 nm. ICE4 was used as the external calibration standard. HPLC analysis was performed with an Agilent 1200 series (Agilent, Santa Clara, CA, USA) chromatography system, and data handling was done using Agilent ChemStation for LC 3D systems (Rev. B. 03.02). The contents of alpha- and beta-acids were reported to two decimal places, calculated based on dry matter. The moisture content was evaluated strictly following the Analytica-EBC 7.2 procedure for moisture content of hops and hop products [30].

### 3.4. Hop Storage Index (HSI)

The HSI value was measured according to the Analytica-EBC protocol for the hop storage index. Ground hops (2.5 g) were combined with 25 mL of toluene and placed on the shaker for 45 min. After extraction, the hops were removed by filtering the extract. Two millilitres of the filtered solution were diluted to 50 mL with methanol (solution A). Solution A was further diluted with alkaline methanol in a 1:24 ratio (*v*/*v*) to produce solution B. The absorbance of solution B was measured spectrophotometrically against a blank sample at 275 nm and 325 nm. The ratio between these wavelengths gave the HSI value [31].

### 3.5. Statistical Analysis

Statistical differences among the different storage conditions, times and hop varieties were analysed using the OriginPro^®^ 2020b (OriginLab Corporation, Northampton, MA, USA) software package.

## 4. Conclusions

This study evaluated the effects of hop storage conditions, variety and form on the hop alpha- and beta- acid content and HSI value. Storage conditions and variety were significant factors for all three parameters. This was the first study mentioning that for beta-acid and HSI values, the pellet form performed better than the cone form. The difference in alpha-acid levels was not statistically significant. The ratio between the alpha- and beta-acids remained constant under anaerobic conditions at 4 °C. Under the other tested storage conditions, an increase in these acids was noted in hop cones and a decrease in the hop pellets, which is important finding, especially for researchers evaluating the influence of old hops on quality of beer bitterness. The percentage of co-humulone in alpha-acids remained stable throughout the entire trial, whereas the percentage of co-lupulone in beta-acids varied. Hop samples stored under anaerobic conditions at 4 °C retained the highest levels of acids and had the lowest HSI values. Therefore, storage in the form of pellets under anaerobic conditions at low temperatures is crucial to maintain the highest possible content of alpha- and beta-acids and the lowest possible HSI and content of oxidation products, thereby providing the desired bitterness in brewed beer. The present study has only investigated the content of alpha-acids and not their oxidation products—humulinones. Further studies would be recommended, since it is known that humulinones have an impact on the formation of beer bitterness when old hops are used in brewing.

## Figures and Tables

**Figure 1 plants-12-00936-f001:**
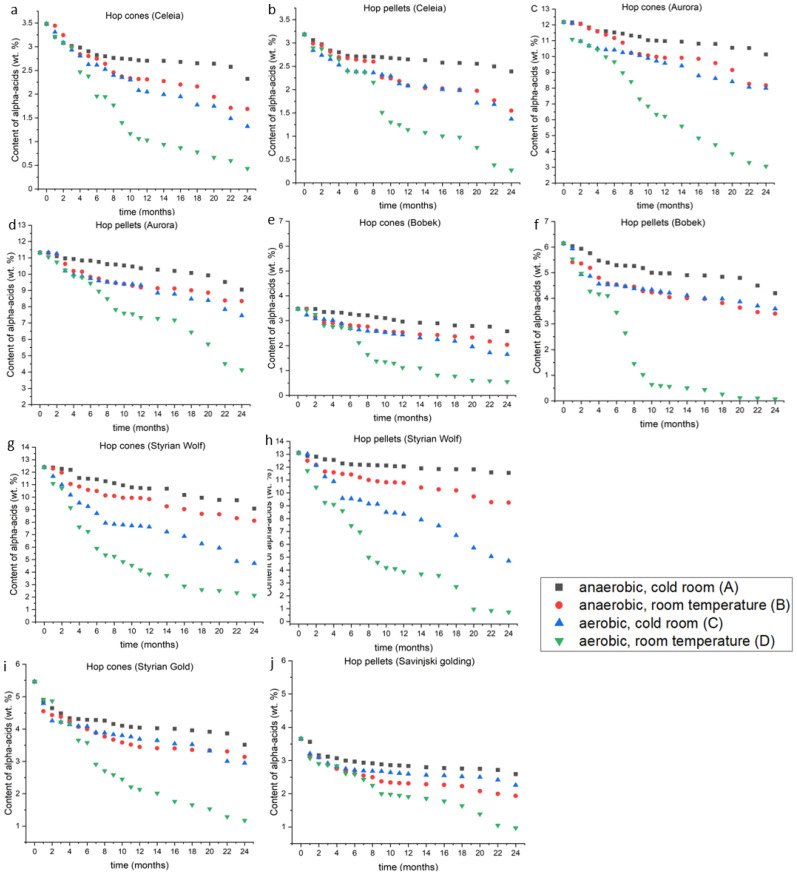
(**a**–**j**): The loss of alpha-acids under different conditions in different varieties during the two-year experiment.

**Figure 2 plants-12-00936-f002:**
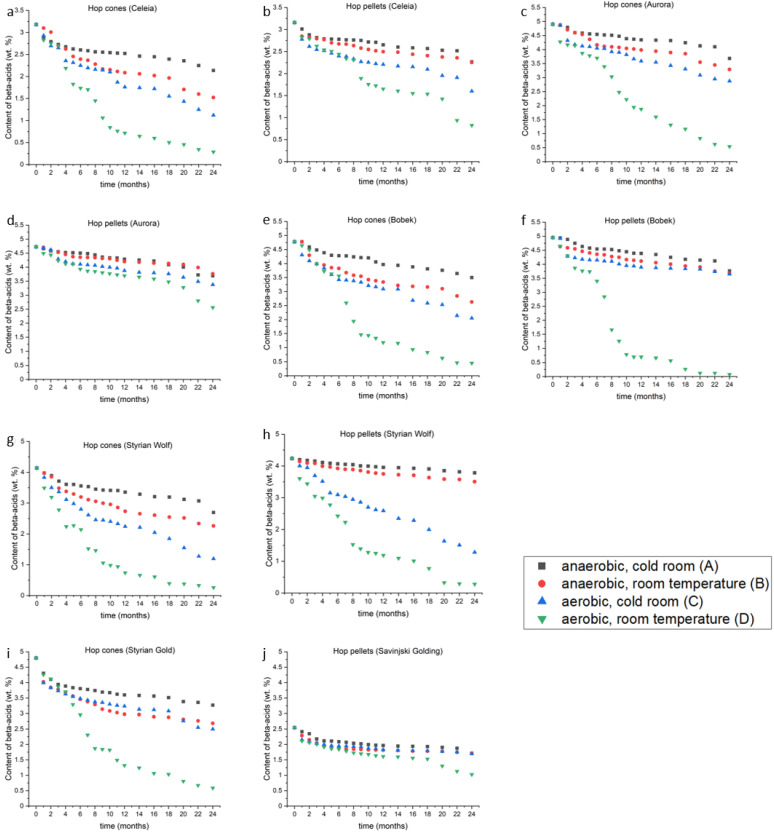
(**a**–**j**): The loss of beta-acids under different conditions in different varieties during the two-year experiment.

**Figure 3 plants-12-00936-f003:**
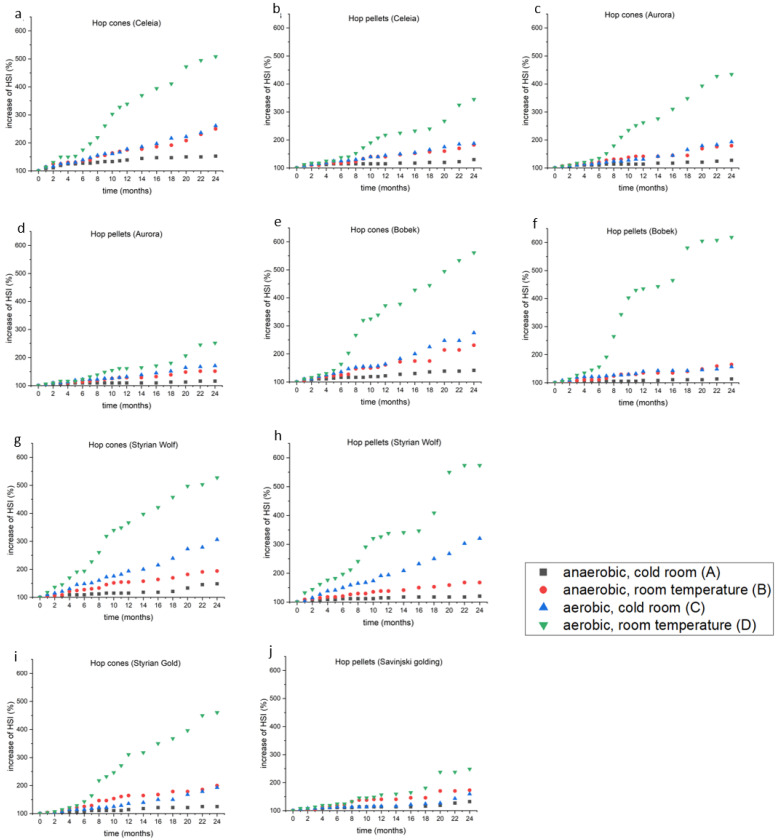
(**a**–**j**): The increase of HSI value under different conditions in different varieties during the two-year experiment.

**Table 1 plants-12-00936-t001:** Losses of alpha-acids at the end of the two year trial.

Storage Conditions	Variety	Shape of Hops	Loss of Alpha-Acids [%]
Anaerobic conditions. cold room	Celeia	cones	33.3 ± 0.8
	Celeia	pellets	24.8 ± 0.6
	Aurora	cones	16.8 ± 0.4
	Aurora	pellets	20.1 ± 0.5
	Bobek	cones	26.1 ± 0.6
	Bobek	pellets	31.7 ± 0.8
	Styrian Wolf	cones	26.8 ± 0.7
	Styrian Wolf	pellets	11.8 ± 0.3
	Styrian Gold	cones	35.5 ± 0.9
	Savinjski golding	pellets	29.0 ± 0.7
Anaerobic conditions. room temperature	Celeia	cones	51.4 ± 1.3
	Celeia	pellets	51.3 ± 1.3
	Aurora	cones	32.9 ± 0.8
	Aurora	pellets	26.2 ± 0.7
	Bobek	cones	41.7 ± 1.0
	Bobek	pellets	44.7 ± 1.1
	Styrian Wolf	cones	34.6 ± 0.9
	Styrian Wolf	pellets	29.5 ± 0.7
	Styrian Gold	cones	42.5 ± 1.0
	Savinjski Golding	pellets	47.0 ± 1.2
Aerobic conditions. cold room	Celeia	cones	62.1 ± 1.5
	Celeia	pellets	52.2 ± 1.4
	Aurora	cones	34.3 ± 0.8
	Aurora	pellets	34.1 ± 0.9
	Bobek	cones	52.6 ± 1.3
	Bobek	pellets	41.6 ± 1.1
	Styrian Wolf	cones	62.1 ± 1.5
	Styrian Wolf	pellets	65.0 ± 1.6
	Styrian Gold	cones	46.0 ± 1.1
	Savinjski Golding	pellets	38.3 ± 1.0
Aerobic conditions. room temperature	Celeia	cones	87.6 ± 2.1
	Celeia	pellets	91.2 ± 2.4
	Aurora	cones	74.9 ± 1.8
	Aurora	pellets	63.5 ± 1.6
	Bobek	cones	84.2 ± 2.1
	Bobek	pellets	98.7 ± 2.5
	Styrian Wolf	cones	82.8 ± 2.0
	Styrian Wolf	pellets	94.4 ± 2.4
	Styrian Gold	cones	78.6 ± 1.9
	Savinjski Golding	pellets	73.5 ± 1.9

**Table 2 plants-12-00936-t002:** Contents of co-humulone, (n+ad)-humulone and % of co-humulone in alpha-acids.

	Time	Co-Humulone				(n+ad)-Humulone				% Co-Humulone in Alpha-Acids			
Variety/ Conditions		A	B	C	D	A	B	C	D	A	B	C	D
Aurora cones	0	2.70 ± 0.05	2.7 ± 0.05	2.7 ± 0.05	2.7 ± 0.05	9.48 ± 0.17	9.48 ± 0.17	9.48 ± 0.17	9.48 ± 0.17	22.2 ± 1.2	22.2 ± 1.2	22.2 ± 1.2	22.2 ± 1.2
	6	2.54 ± 0.04	2.43 ± 0.04	2.29 ± 0.04	2.16 ± 0.04	8.98 ± 0.16	8.75 ± 0.16	8.12 ± 0.15	7.5 ± 0.13	22.1 ± 1.2	22.0 ± 1.1	22.4 ± 1.1	22.4 ± 1.1
	12	2.43 ± 0.04	2.2 ± 0.04	2.13 ± 0.04	1.39 ± 0.02	8.55 ± 0.15	7.72 ± 0.14	7.44 ± 0.13	4.83 ± 0.09	22.1 ± 1.1	22.3 ± 1.1	22.3 ± 1.1	22.6 ± 0.9
	18	2.39 ± 0.04	2.11 ± 0.04	1.95 ± 0.03	1.03 ± 0.02	8.41 ± 0.15	7.47 ± 0.13	6.66 ± 0.12	3.43 ± 0.06	22.1 ± 1.1	22.6 ± 1.1	23.1 ± 1.1	22.7 ± 0.8
	24	2.26 ± 0.04	1.79 ± 0.03	1.75 ± 0.03	0.70 ± 0.01	7.88 ± 0.14	6.38 ± 0.11	6.25 ± 0.11	2.35 ± 0.04	22.3 ± 1.1	21.9 ± 1.0	23.0 ± 1.0	22.9 ± 0.7
Styrian Wolf cones	0	2.69 ± 0.05	2.69 ± 0.05	2.69 ± 0.05	2.69 ± 0.05	9.71 ± 0.17	9.71 ± 0.17	9.71 ± 0.17	9.71 ± 0.17	21.7 ± 1.2	21.7 ± 1.2	21.7 ± 1.2	21.7 ± 1.2
	6	2.47 ± 0.04	2.32 ± 0.04	1.92 ± 0.03	1.43 ± 0.02	8.96 ± 0.16	8.16 ± 0.15	6.77 ± 0.12	5.02 ± 0.09	21.6 ± 1.1	22.2 ± 1.1	22.1 ± 1.0	22.2 ± 0.9
	12	2.34 ± 0.04	2.13 ± 0.04	1.68 ± 0.03	0.88 ± 0.02	8.35 ± 0.15	7.71 ± 0.14	5.94 ± 0.11	2.96 ± 0.05	21.9 ± 1.1	21.7 ± 1.1	22.0 ± 1.0	22.9 ± 0.8
	18	2.23 ± 0.04	1.89 ± 0.03	1.38 ± 0.02	0.58 ± 0.01	7.72 ± 0.14	6.77 ± 0.12	4.9 ± 0.09	2.01 ± 0.04	22.4 ± 1.1	21.9 ± 1.0	22.0 ± 0.9	22.3 ± 0.6
	24	1.99 ± 0.03	1.79 ± 0.03	1.04 ± 0.02	0.45 ± 0.01	7.09 ± 0.13	6.32 ± 0.11	3.66 ± 0.07	1.68 ± 0.03	21.9 ± 1.0	22.1 ± 1.0	22.2 ± 0.8	21.2 ± 0.6
Aurora pellets	0	2.55 ± 0.04	2.55 ± 0.04	2.55 ± 0.04	2.55 ± 0.04	8.77 ± 0.16	8.77 ± 0.16	8.77 ± 0.16	8.77 ± 0.16	22.5 ± 1.2	22.5 ± 1.2	22.5 ± 1.2	22.5 ± 1.2
	6	2.42 ± 0.04	2.22 ± 0.04	2.21 ± 0.04	2.13 ± 0.04	8.41 ± 0.15	7.61 ± 0.14	7.53 ± 0.13	7.31 ± 0.13	22.4 ± 1.1	22.6 ± 1.1	22.7 ± 1.1	22.5 ± 1.1
	12	2.35 ± 0.04	2.08 ± 0.04	2.1 ± 0.04	1.67 ± 0.03	8.02 ± 0.14	7.11 ± 0.13	7.22 ± 0.13	5.66 ± 0.10	22.6 ± 1.1	22.6 ± 1.1	22.5 ± 1.1	22.8 ± 1.0
	18	2.29 ± 0.04	2.01 ± 0.03	1.94 ± 0.03	1.45 ± 0.03	7.78 ± 0.14	6.99 ± 0.13	6.54 ± 0.12	4.99 ± 0.09	22.7 ± 1.1	22.3 ± 1.0	22.9 ± 1.0	22.5 ± 0.9
	24	2.07 ± 0.04	1.86 ± 0.03	1.70 ± 0.03	0.91 ± 0.02	6.98 ± 0.12	6.49 ± 0.12	5.76 ± 0.1	3.22 ± 0.06	22.9 ± 1.1	22.3 ± 1.0	22.8 ± 1.0	22.0 ± 0.8
Styrian Wolf pellets	0	3.01 ± 0.05	3.01 ± 0.05	3.01 ± 0.05	3.01 ± 0.05	10.1 ± 0.18	10.1 ± 0.18	10.1 ± 0.18	10.1 ± 0.18	23.0 ± 1.3	23.0 ± 1.3	23.0 ± 1.3	23.0 ± 1.3
	6	2.78 ± 0.05	2.61 ± 0.05	2.17 ± 0.04	1.70 ± 0.03	9.44 ± 0.17	8.83 ± 0.16	7.38 ± 0.13	5.74 ± 0.10	22.7 ± 1.2	22.8 ± 1.2	22.8 ± 1.1	22.9 ± 1.0
	12	2.74 ± 0.05	2.43 ± 0.04	1.91 ± 0.03	0.87 ± 0.02	9.31 ± 0.17	8.34 ± 0.15	6.44 ± 0.12	3.00 ± 0.05	22.7 ± 1.2	22.6 ± 1.1	22.9 ± 1.0	22.4 ± 0.7
	18	2.71 ± 0.05	2.31 ± 0.04	1.54 ± 0.03	0.60 ± 0.01	9.13 ± 0.16	7.89 ± 0.14	5.16 ± 0.09	2.10 ± 0.04	22.9 ± 1.2	22.6 ± 1.1	23.0 ± 0.9	22.2 ± 0.7
	24	2.62 ± 0.05	2.09 ± 0.04	1.05 ± 0.02	0.16 ± 0.01	8.94 ± 0.16	7.15 ± 0.13	3.55 ± 0.06	0.56 ± 0.01	22.7 ± 1.2	22.6 ± 1.1	22.8 ± 0.8	22.4 ± 0.5

**Table 3 plants-12-00936-t003:** Change of alpha/beta ratio at the beginning and the end of the storage trial.

Alpha/Beta Ratio									
		Conditions							
Variety	Form	A (Beginning)	A (End)	B (Beginning)	B (End)	C (Beginning)	C (End)	D (Beginning)	D (End)
Celeia	cones	1.1	1.1	1.1	1.1	1.1	1.2	1.1	1.5
	pellets	1.0	1.1	1.0	0.7	1.0	0.9	1.0	0.3
Aurora	cones	2.5	2.8	2.5	2.5	2.5	2.8	2.5	5.7
	pellets	2.4	2.5	2.4	2.2	2.4	2.2	2.4	1.6
Bobek	cones	0.7	0.7	0.7	0.8	0.7	0.8	0.7	1.2
	pellets	1.2	1.1	1.2	0.9	1.2	1.0	1.2	1.0
Styrian Wolf	cones	3.0	3.4	3.0	3.6	3.0	3.9	3.0	8.2
	pellets	3.1	3.1	3.1	2.6	3.1	3.6	3.1	2.6
Styrian Gold	cones	1.1	1.1	1.1	1.2	1.1	1.2	1.1	2.0
Savinjski golding	pellets	1.4	1.5	1.4	1.1	1.4	1.3	1.4	0.9

**Table 4 plants-12-00936-t004:** Correlation coefficients between the loss of acids and HSI.

	Correlation Coefficient R^2^									
Conditions/ Variety	Celeia		Aurora		Bobek		Styrian Wolf		Styrian Gold/Savinjski Golding	
	Cones	Pellets	Cones	Pellets	Cones	Pellets	Cones	Pellets	Cones	Pellets
A	0.96	0.95	0.97	0.89	0.97	0.93	0.88	0.88	0.88	0.88
B	0.95	0.97	0.98	0.94	0.92	0.94	0.91	0.87	0.91	0.87
C	0.97	0.96	0.94	0.95	0.96	0.91	0.88	0.74	0.88	0.74
D	0.93	0.97	0.98	0.98	0.96	0.94	0.93	0.92	0.93	0.92

## Data Availability

Data is contained within the article or Appendix A.

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
