# Peer review of "The Stability of Hop (Humulus lupulus L.) Resins during Long-Period Storage"

_plants, 2023, doi:10.3390/plants12040936_

Round 1
Reviewer 1 Report
The article entitled “The Stability of Hop (Humulus lupulus L.) Resins during Long-Period Storage”, submitted to the journal presents a two-year study of the dynamics of loss of hop alpha and beta acids during hop storage in different conditions. Hop quality losses during storage are still in the spotlight, hops of a certain variety are traded primarily on the basis of their alpha acid content.
The manuscript is clearly structured and written in a language understandable to potential readers. The range of analyses used is suitable for the declared aim of the study. The results were statistically processed and are mostly satisfactorily discussed.
I have the following comments on the article:
- Comparison of changes in hop pellets and cones during storage is one of the stated goals, for better orientation of the reader I recommend listing varieties tested in only one form at the end in tables and figures.
- Based on current knowledge, it is quite likely that the oxidation products of alpha acids (humulinones) and especially beta acids (hulupones) are significantly manifested in the bitterness of beers only during dry hopping. This should be mentioned in the discussion, especially in chapter 3.2.
- The discussion of a number of factors is difficult, there are minor inaccuracies in the text. I recommend that the authors go through the manuscript carefully again.
- The conclusion should answer the aim of this study. The conclusion should better capture what new insights have been found and how these insights advance our knowledge in the field.
On my opinion, the work brought a number of findings for further research and practice and manuscript could be accepted for the publication after minor revision.
Author Response
Reviewer: 1
The article entitled “The Stability of Hop (Humulus lupulus L.) Resins during Long-Period Storage”, submitted to the journal presents a two-year study of the dynamics of loss of hop alpha and beta acids during hop storage in different conditions. Hop quality losses during storage are still in the spotlight, hops of a certain variety are traded primarily on the basis of their alpha acid content.
The manuscript is clearly structured and written in a language understandable to potential readers. The range of analyses used is suitable for the declared aim of the study. The results were statistically processed and are mostly satisfactorily discussed.
- Comparison of changes in hop pellets and cones during storage is one of the stated goals, for better orientation of the reader I recommend listing varieties tested in only one form at the end in tables and figures.
As suggested, we moved Styrian Gold cones and pellets of Savinjski golding at the end of all tables and figures.
- Based on current knowledge, it is quite likely that the oxidation products of alpha acids (humulinones) and especially beta acids (hulupones) are significantly manifested in the bitterness of beers only during dry hopping. This should be mentioned in the discussion, especially in chapter 3.2
We added information on humulinones and hulupones contribution to beer bitterness in chapter 3.2
- The discussion of a number of factors is difficult, there are minor inaccuracies in the text. I recommend that the authors go through the manuscript carefully again.
We have re-read the whole article again and eliminated some minor inaccuracies.
- The conclusion should answer the aim of this study. The conclusion should better capture what new insights have been found and how these insights advance our knowledge in the field.
The conclusion was re-written in order to better capture the new insights found.

Reviewer 2 Report
"Introduction" section is too long, I recommend to make it more concise.
Author Response
Reviewer: 2
- "Introduction" section is too long, I recommend to make it more concise.
We have shorten the introduction section.

Reviewer 3 Report
The bitter acid content of hops is important for beer brewing. However, in industrial production, the iso-α-acid content is the reference for hop addition. Does the author focus on the iso-α-acid? Meanwhile, the humulinone is another oxidative products of humulinone, which also contributs to the bitter taste.
Author Response
Reviewer: 3
- The bitter acid content of hops is important for beer brewing. However, in industrial production, the iso-α-acid content is the reference for hop addition. Does the author focus on the iso-α-acid? Meanwhile, the humulinone is another oxidative products of humulinone, which also contributs to the bitter taste.
Iso-alpha-acids are isomerisation products of the alpha-acids. For their isomerisation high temperature is needed (higher than room temperature), meaning that iso-alpha-acids are produced during boiling the hop wort and are not found in the hop as themselves. There are some minor contents of iso-alpha-acids in hops as a consequences of kilning, but they are not products of hop aging. So the focus were on alpha-acids, source of iso-alpha-acids. We agree that humulinones are oxidative products of humulones, but their content was not analysed in the experiment.

Round 2
Reviewer 3 Report
The stability of bitter acid is important for hops. Oxidation of hops occured at room temperature. In beer, especially in IPA, the humulinones occupied a significant ratio of bitter acids (~50% ). I think it's important to figure out the contents of humulinones. If you have not studied in this experiment, it's better to note it in conclusion.
Author Response
Reviewer: 3
- The stability of bitter acid is important for hops. Oxidation of hops occured at room temperature. In beer, especially in IPA, the humulinones occupied a significant ratio of bitter acids (~50% ). I think it's important to figure out the contents of humulinones. If you have not studied in this experiment, it's better to note it in conclusion.
We agree that humulinones are very important compounds in hop aging and consequentially in formation of hop bitterness. Unfortunately we have not measured them during two year experiment. We have included notation about their importance and the recommendation for further studies about them into the conclusion.
